

# The Standardized Vertical Velocity Anomaly Index (SVVAI): Using Atmospheric Dynamical Anomalies to Simulate and Predict Meteorological Droughts

Zhenchen Liu[1], Hai He[1], Zhiyong Wu[1], Guihua Lu[1], and Hao Yin[1]

[1]Institute of Water Problem, College of Hydrology and Water Resources, Hohai University, Nanjing, 210098, China

*Correspondence to*: Hai He (hehai_hhu@hhu.edu.cn)

**Abstract.** Vertically downward motion of air current is physically drought-inducing, which has the potential of being a simple and universal drought indicator. The core objective of the present study is to employ vertical motion to simulate and

predict droughts after investigating dynamically drought-inducing mechanism. Season-scale drought processes and spatial distributions during 2009–2016 are our concerns, and all the drought study regions of China were chosen as the research areas. Three-month SPI (SPI3) updated daily was used to identify drought processes, and original vertical motion and associated horizontal divergence were also transformed to season-scale standardized anomalies (SA) with a daily running window. *In situ* observation, ERA-Interim reanalysis, and CFSv2 forecast products were comprehensively employed for

drought simulation and prediction. To date, the main results and conclusions are as follow: (1) Atmospheric dynamical anomalies during drought processes and key phases were uncovered. Dynamically drought-inducing features are generally characterized as the typically anomalous "upper-convergence–lower-divergence" patterns and the intensified downward vertical motion as expected. Signal intensities and vertical configurations are time-varying and seemingly coincide with evolution of regional processes. Particularly, vertical velocity exhibited universally strengthened downward anomalies over

almost all the droughts. (2) On the basis of dynamically vertical features uncovered above, the SVVAI (Standardized Vertical Velocity Anomaly Index) is newly proposed. The SVVAI is calculated using SA-based values of vertical motion at multiple pressure levels in the troposphere. The SVVAI_ave and SVVAI_max, corresponding to the vertically average- and maximum-based computation schemes, can be adopted. (3) Drought processes and spatial distributions were simulated with the SVVAI_ave and SVVAI_max. They commonly show highly positive correlations with realistic ones over most regions,

and the SVVAI_ave outperformed the SVVAI_max. (4) To further understand difference of simulation capacity, temporal correlation coefficients (TCC) of the SVVAI_ave against observed SPI3 at the grid scale were used for analysis. Positive TCCs above +0.3 occupies most areas to the east of 110º E, while large-area low TCCs (-0.1～+0.3) appear to the west of 110º E over China. It is notably seen that East China and Northeast China are the two regions with highly positive TCCs (+0.6～+0.8). (5) Drought prediction using the SVVAI_ave was preliminarily explored. Regarding the prospective 60-day

process prediction, the predicted SVVAI_ave was equally matched with or a little better than the forecasted SPI3 in most



cases. Predicted spatial distribution is preliminarily assessed via the example of the 2011 summer-autumn drought over Southwest China, and prediction performance at the occurrence, peak and termination times are inconsistent. (6) Overall, the novel SVVAI herein may be complementary to current approaches of operational drought monitoring and prediction. Further study could be focused on the two following aspects: One is index applicability, that is to say, to explore when and where the

predicted SVVAI outperforms the forecasted SPI3. The other is to further explore antecedent drought-inducing signals of atmospheric/oceanic anomalies with the bridge of vertical motion, which may provide a fundamental approach for drought prediction with long lead times.

## 1 Introduction

Drought is an economically and ecologically disruptive natural hazard that profoundly impacts water resources, agriculture, ecosystems, and basic human welfare (Dai, 2011). In the latest decade, extremely severe drought events hit many regions in the world (Funk, 2011;Seager et al., 2015;Ionita et al., 2017;Wang et al., 2015), and reliable drought prediction is therefore fundamental for water resource managers to develop and implement mitigation measures. Multiple approaches of drought prediction and relevant improvement measures have been comprehensively investigated in the hydro-climate research

community (Hao et al., 2018). Because lack of precipitation is the root cause of droughts, meteorological drought on the basis of long-term precipitation deficit is one of the focuses. By far, much attention has been mainly paid to prediction of meteorological drought, and methods include statistical prediction(Hao et al., 2014;Hao et al., 2016;Pan et al., 2013;Costa-Cabral et al., 2016), dynamical forecasting (Ma et al., 2015;Yuan and Wood, 2013;Yuan et al., 2015), and hybrid statistical-dynamical prediction(Tian et al., 2014;Madadgar et al., 2016;Wang et al., 2017;Xu et al., 2018). Regardless of approach

differences, they commonly rely on precipitation products derived from climate forecast systems (Saha et al., 2014;Kirtman et al., 2014), sometimes with the help of some typically inherent teleconnection-related climate indices (e.g., the North Atlantic Oscillation (NAO) and El Niño–Southern Oscillation (ENSO))(Wang et al., 2017;Moreira et al., 2016;Bonaccorso et al., 2015).

As a valuable attempt, physically drought-inducing causes are treated or generalized as indicators of drought prediction.

Some atmospheric horizontal circulation patterns (i.e., weather regimes), which are closely related to temperature- and moisture-related advection processes of large-scale atmospheric flows, indirectly affect region-scale drought development (Kingston et al., 2015;Fowler and Kilsby, 2002). Similar to well recognized teleconnection patterns (Wallace and Gutzler, 1981;Hurrell, 1995), they are target-oriented and conceptually extended, which were considered in some physically conceptual models for drought prediction (Layaysse et al., 2018;Liu et al., 2018;Richardson et al., 2018). For example,

Lavaysse et al. (2018) used 30-day-lead forecasted monthly occurrence anomalies of weather regimes to predict one-month-scale meteorological drought over Europe, which helps detect ~65% of droughts in winter in northeastern European continent one month in advance. At the same time, Liu et al. (2018) developed a conceptual prediction model of regional



drought processes forced with forecasted atmospheric variables, which calibrated the statistically concurrent relationship between anomalies mainly at 200/500 hPa geopotential height fields and season-scale meteorological droughts, and model
application showed its good performance in predicting process development.

In comparison with quasi-horizontal motions featuring circulation patterns or weather regimes mentioned above, vertical motion is another potential drought indicator, which more directly affect drought development by means of controlling the upward or downward motion of converged water vapour. Actually, there are considerable researches to explore the roles of vertical motion in drought formation. On one hand, as one direct drought-inducing cause, anomalous vertical motion could
help understand drought mechanism (Wang et al., 2017;Zeng et al., 2019b), which was usually diagnosed as one anomalous branch of atmospheric circulation in the form of vertical–meridional cross sections. On the other hand, vertical motion is also an important bridge linking large-scale oceanic-atmospheric circulation to strong droughts or region-scale rainfall variability (Roucou et al., 1996;Long et al., 2000;Paredes Trejo et al., 2016). For example, Long et al. (2000) employed vertical wind velocity at 500 hPa to investigate the role of large-scale general circulation features in modulating rainfall variability over
sub-Saharan regions, and results suggest that general circulation were important in initiating drought in the Sahel.

These preliminary achievements encouraged us to explore the further application of vertical motion in drought simulation and prediction. Liu et al. (2017) previously employed anomalies of horizontal divergence that is closely related to vertical motion in nature to build a dynamical index, which highly correlated with season-scale pluvial-drought process transition over Southwest China. Besides, the motivation of the present study also originates from the following points: Firstly, as one
physically important causes of meteorological droughts, vertical motion seems to have the potential of being a universal drought indicator, regardless of geographical difference and seasonality. Secondly, with respect to the computation procedures of climate forecast systems, three-dimensional wind fields is calculated before precipitation rate. Since numerical computation errors generally tend to accumulate with advance of the computation progress, a vertical motion-based index forced by the same climate forecast products might perform better on drought prediction than a precipitation-based drought
index does.

Accordingly, the main objective of this study is to employ vertical motion to simulate and predict drought processes and spatial distributions. The corresponding fundamental issues are as follow: (1) Vertical structures and temporal evolution of vertical motion and associated horizontal divergence were investigated in Sect. 3, which help uncover features of atmospheric dynamical anomalies during region-scale drought processes and key phases. (2) Based on knowledge of
dynamically drought-inducing mechanism in Sect. 3, the SVVAI (Standardized Vertical Velocity Anomaly Index) calculated using anomalous values of vertical motion in the troposphere was newly proposed in the study. Further, the SVVAI_ave and SVVAI_max, corresponding to the average- and maximum-based computation schemes of the SVVAI, are taken as new drought indicators for simulation of processes and spatial patterns in Sect. 4. (3) Further, drought prediction using the SVVAI_ave was preliminarily explored in Sect. 6. Additionally, methodology, discussion, and conclusion parts can be found
in Sect. 2, 7, and 8, respectively.





## 2 Methodology

### 2.1 Study areas

China has a wide range of landform features, including flatlands over eastern China, grasslands over Inner Mongolia, highlands over Tibet and deserts over Xinjiang. As such, China is a typically comprehensive example for drought-related

analysis. Since landform difference leads to consideration of drought regionalization, nine drought study regions over China previously employed (Wu et al., 2011) are also the focus in the present study. In addition, spatial distribution of surface pressures reflect elevation variations over China (Fig. 1 (b)), which is a vitally constraint factor of the dynamically based drought index newly proposed in the present study.

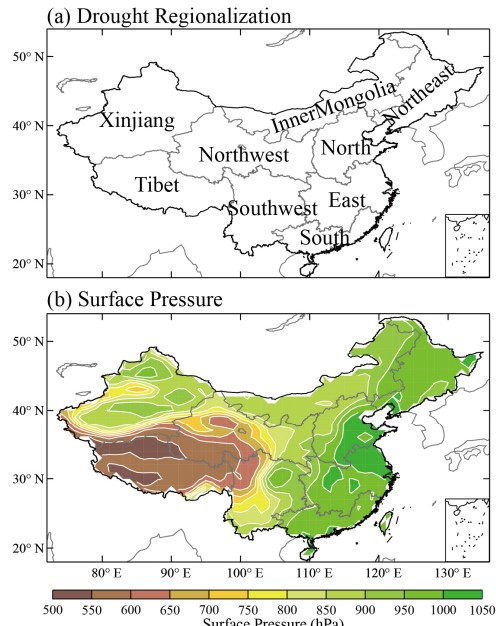


**Figure 1.** Drought regionalization (Wu et al., 2011) (a) and spatial distributions of multi-year averaged surface pressure during 1981–2016 (b) over China





**2.2 Data sources**

**2.2.1 Dynamically atmospheric variables**

Both vertical velocity (hereafter simplified as ω) and associated horizontal divergence of the realistic three-dimensional wind fields are the two dynamically atmospheric variables potential for drought simulation and prediction in the present study. For drought simulation, they could be retrieved from the ERA-Interim reanalysis data set (Dee et al., 2011); Regarding drought prediction, forecasted vertical velocity sources from the NCEP Climate Forecast System Version 2 (CFSv2)(Saha et al.,

2014). Details about both vertical velocity and horizontal divergence, shared by the ERA-Interim reanalysis and CFSv2 forecasted data sets, are listed in Table 1.

**Table 1.** Detailed information of two dynamically atmospheric variables employed in the present study

| Variable | Unit | Abbreviation | Pressure level | Spatial resolution | |
| --- | --- | --- | --- | --- | --- |
| | | | | Horizontal | Vertical |
| Horizontal divergence | 1/s | - | 100–1000 hPa | 1°×1° | 100 to 250 hPa by 25 hPa, 300 to 750 hPa by 50 hPa, and 775 to 1000 hPa by 25 hPa |
| Vertical velocity | Pa/s | ω | | | |

Specifically speaking, the ERA-Interim reanalysis data sets provided by the European Centre for Medium-Range Weather Forecasts (ECMWF) were employed for drought simulation herein. The 6-hourly (UTC 00, 06, 12, and 18) reanalysis data subsets from 1 October 1980 to 31 December 2016 were chosen and transformed to a daily time scale with the simple time-weighted mean method. In addition, spatial distributions of surface pressures over China were also calculated on the base of ERA-interim reanalysis data with a resolution of 1°×1°. The 6-hourly surface pressure data set was also transformed to a

daily time scale, and multi-year season-scale average values were firstly calculated. Mainly due to slight seasonality, maps of multi-year spatial distribution in different seasons were simply averaged as shown in Fig. 1 (b).

CFSv2 products were used to verify performance of drought prediction using the newly proposed ω based drought index. Initialized at each ten days, the prospective 60-day forecasted time series during 2009–2016 were retrieved and eventually used for process prediction. Details about extraction of CFSv2 forecasted datasets can be found in Sect. S1 of the supplement

file.

**2.2.2 Precipitation data**

The long-term observed precipitation data set over China was employed for calculation of precipitation based drought indices, whose source is the second-version Dataset of Observed Daily Precipitation Amounts at each 0.5°×0.5° grid point in

China for 1981–2016 (http://data.cma.cn/data/detail/dataCode/SURF_CLI_CHN_PRE_DAY_GRID_0.5.html) kindly provided by China Meteorological Administration (CMA). They were transformed to daily gridded data set with a resolution of 1°×1° using the nearest neighbour interpolation method. Following the aforementioned drought regionalization (Fig. 1 (a)),





daily area-averaged precipitation amounts over nine drought regions in China were also computed. In addition, forecasted precipitation data could also be retrieved from CFSv2 forecast subsets, which were also used to compute the forecasted

precipitation-based drought index for inter-comparison with the novel dynamically-based drought index.

### 2.3 Identification of regional drought processes and key period division

Regional severe and extreme drought processes, together with key phases of occurrence, persistence, peak and recovery, are the focus in the present study. Key methods and steps are as follow:

**2.3.1 Three-month SPI (SPI3) updated daily**

Three-month SPI (McKee and Kleist, 1993), hereafter simplified as SPI3, was employed as the precipitation-based drought index to identify season-scale drought. In order to analysis temporal variation of drought processes, we chose to update SPI3 daily. The period for computation is from 1 January 1981 to 31 December 2016, and other details can be found in Sect. 3.1 of our previous study (Liu et al., 2018).

**2.3.2 Identification of severe drought processes**

On the basis of daily region-scale SPI3 time series during 1981–2016, drought processes are identified when the daily SPI3 values are below -0.50 for more than 60 consecutive days. Details about process identification could be found in Sect. 3.2 of our previous study (Liu et al., 2018), while the critical proportion was set to 25% in the present study. Eventually, only those extreme and severe processes are our concern.

**2.3.3 Key Phase Division**

During one complete drought process, periods between the first and last day with SPI3 below -1.5 are referred to as drought persistence, and the item "persistence" is usually labelled on its first day. Similarly, the last day with SPI3 below -1.5 corresponds to the start date of drought recovery. In addition, the minimum SPI3 value during one process represents drought peak.

Following the aforementioned steps, severe or extreme regional drought processes during 1981–2016, together with key phases, were identified and divided. In the present study, eighteen processes during 2009–2016 are the focus, and detailed information can be found in Table S1 of the supplement file. For the sake of brevity and readability, only six typical region-scale processes are analysed in detail in the manuscript, while results of the other processes are provided in the supplement file.




### 2.4 Physical meanings of dynamic variables and their forms of Standardized Anomalies (SA)

Both horizontal divergence and vertical velocity, which are potential for drought-inducing analysis and construction of new drought indices, have explicitly physical meanings. Following ECMWF parameter description (https://apps.ecmwf.int/codes/grib/param-db/?id=155), horizontal divergence of the three-dimensional wind fields is the rate

at which airflow is spreading out horizontally from a point, and the positive (negative) values correspond to divergence (convergence) of airflow. Vertical velocity (ω) is the speed of airflow motion in the upward or downward direction. Negative (positive) values of vertical velocity indicate upward (downward) motion, chiefly due to the pressure based vertical coordinate system and the natural law of pressure decreasing with height.

Specifically speaking, vertical velocity (ω) is physically related to horizontal divergence, which could be easily understood

via the continuity equation in the field of dynamic meteorology (see section 2.5 and section 3.5 in the book of dynamic meteorology(Holton, 2004)). In practice, the calculation of vertical velocity is complex, which output from numerically complex processes rather than simply obtain from being minus horizontal divergence.

Consistent with the time scale and none-dimensional form of SPI3, these two dynamic variables were also transformed to season-scale Standardized Anomalies (SA). Similar to application in drought simulation and prediction previously (Liu et al.,

2017, 2018), The season-scale SA of the two dynamic variables is defined (Hart and Grumm, 2001) following Eq. (1):

$$SA = \frac{X-\mu}{\sigma}, \tag{1}$$

where SA is the Standardized Anomalies at both grid and region scales, X denotes the original meteorological variable, and $\mu$ and $\sigma$ are the mean value and the standard deviation. In the present study, X is the originally dynamic variable defined as the 90-day-mean value located on the last day updated daily. Accordingly, both $\mu$ and $\sigma$ for the climatologically 1981–2016

period are calculated based on these 90-day-mean values.

### 2.5 Definition and computation of Standardized Vertical Velocity Anomaly Index (SVVAI)

To quantitatively apply vertical velocity (ω) in drought simulation and prediction, the so-called SVVAI (Standardized Vertical Velocity Anomaly Index) is proposed. The SVVAI is conceptually the index calculated based on SA of vertical velocity in the troposphere. In order to compute it, the simply average and max based schemes were adopted. The

computation scheme of both SVVAI_ave and SVVAI_max follow Eq. (2) and (3):

$$SVVAI\_ave = -\text{average}(\textstyle\sum_{i=\min(850\,\text{hPa},P_{surface})}^{200\,\text{hPa}} \omega SA90), \tag{2}$$

$$SVVAI\_max = -\max(\omega SA90_{\min(850\,\text{hPa},P_{surface})}, \cdots, \omega SA90_{200\,\text{hPa}}), \tag{3}$$

where ωSA90 is season-scale (90-day in practice) SA of vertical velocity (ω) at one certain level within the troposphere. In general, the pressure levels of 200 hPa and 850 hPa are usually chosen for analyse on horizontal convergence and divergence

of large-scale wind fields, which are physically related to vertical motions in the troposphere. Accordingly, the upper



limitation for computing ωSA90 reaches 200 hPa, whereas the lower limitation is designed to be the minimum pressure value between 850 hPa and surface pressure P$_{surface}$ mainly due to relatively low values near surface pressure to the west of 100$^{o}$ E over China (Fig. 1 (b)).

**2.6 Drought simulation, prediction and assessment**

The SVVAI_max and SVVAI_ave can be simply calculated only using ERA-Interim reanalysis datasets for drought simulation, whereas the prospective 60-day drought prediction are the focus. The SVVAI itself corresponds to temporally concurrent SPI3, and therefore it has no lead time. Their lead time depends on that of climate forecast models (e.g., CFSv2). In practice, both SVVAI_ave and SVVAI_max were operationally forced with a combination of CFSv2 forecast products and ERA-Interim reanalysis datasets. Detailed procedures and illustrations can be found in Fig. S1 of the supplement file.

Temporal correlation coefficients (TCC) and pattern correlation coefficients (PCC) are employed for assessment of simulation and prediction, as also used in other drought-related analysis (Pu et al., 2016) and seasonal forecast verification (Yu et al., 2018). Essentially, both TCC and PCC are the "Pearson product-moment coefficient of linear correlation" between two variables x and y, i.e., the term Correlation Coefficients (CC), following Eq. (4):

$$CC = \frac{\sum_{k=1}^{m}(x_k - \bar{x})(y_k - \bar{y})}{\sqrt{\sum_{k=1}^{m}(x_k - \bar{x})^2 \sum_{k=1}^{m}(y_k - \bar{y})^2}},$$  (4)

Here x and y denote variables regarding time series or spatial fields, while $\bar{x}$ and $\bar{y}$ are the average values over a given M-point time series or a map of M grid-points. Relevant details can be found in Sect. S2 of the supplement file.

**3 Vertical structures and temporal evolutions of atmospheric dynamical anomalies**

Vertical velocity and associated horizontal divergence are employed to investigate the role of atmospheric dynamical anomalies in evolution of drought processes (Fig. 2 and Fig. S2). As commonly revealed in Fig. 2, negative divergence SA
almost located above positive divergence SA and accompanied with positive ω SA between them, dynamically characterized as the typically anomalous "upper-convergence–lower-divergence" pattern and the intensified downward vertical motion during drought processes. Physically, the anomalously converged horizontal wind fields in the upper levels could diverge in the lower levels, and accordingly intensive downward anomalies appear along the vertical direction.



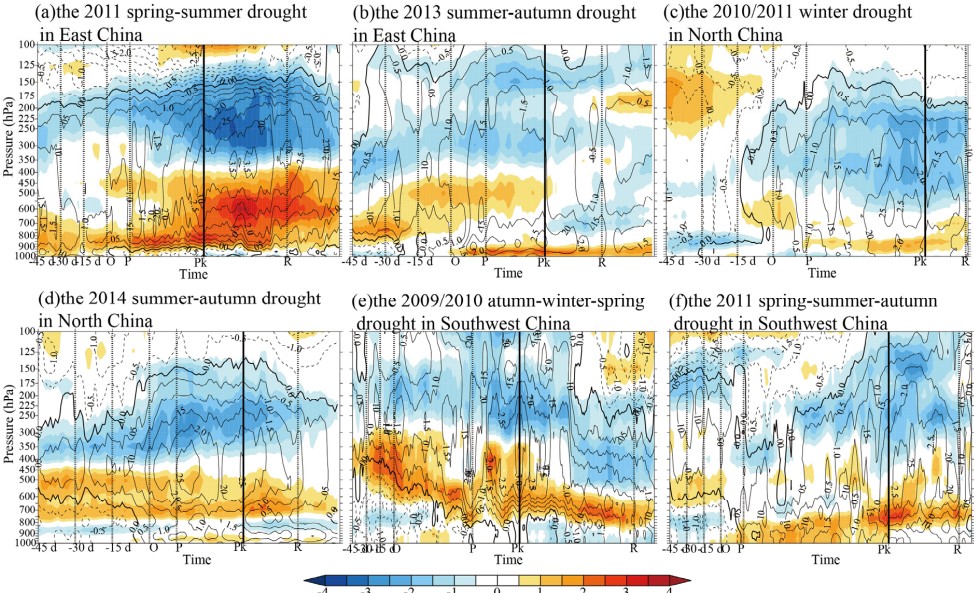

**Figure 2.** Temporal evolution of Standardized Anomalies (SA) of season-scale horizontal divergence (shadow) and vertical velocity (contours) regarding six typical region-scale drought processes during 1981–2016 over China. The interval for shadow and contours is 0.5, and solid (dashed) contours mean positive (negative) values. The occurrence, persistence, peak and recovery phases are referred to as the labels "O", "P", "Pk", and "R", respectively. The labels "-45 d", "-30 d" and "-15 d" correspond to the previous days before drought occurrence.

It is notable that dynamically vertical structures (e.g., signal intensity and relative position) are time-varying and seem to coincide with regional drought processes when evolved from occurrence to termination. Anomalous intensities of both vertical velocity and horizontal divergence reach almost maximum values during drought persistence (usually near the time of drought peak) and are relatively weak at the phases of drought occurrence or recovery.

To further clarify the aforementioned temporal features by stages, statistical modes of SA-based samples during each process and stages are plotted in Fig. 3 and Fig. S3. Even with some difference, the vertically "upper-convergence–lower-divergence" structure and associated strongly downward motion over almost all the processes can be commonly found as depicted by the thick black curves, and the 2011 East China drought is taken as a typical example for detailed analysis. Evolved from occurrence to persistence (Fig. 3 (a)), the "upper-convergence–lower-divergence" structure clearly appeared between 200 hPa and 850 hPa and the divergence SA values dramatically increased, and accordingly vertically peak values of ω SA increased from +1.0 to +3.5. However, when evolved from persistence to recovery, the divergence-SA-based vertical structure and the positive ω SA got weakened, rather than the pattern-reversed structure or the obviously upward motion.


It is not always the explicit case with a clearly vertical structure. Firstly, vertical structures and evolution by stages during the 2009/2010 and the 2011 droughts in Southwest China are not as orderly as those in East China and North China (Fig. 3),
easily explained by the intensively strength-varying and position-varying divergence SA structure (Fig. 2 (e) and (f)). Secondly, the divergence-SA-based vertical structure in other regional drought processes could exhibit two or more "upper-convergence–lower-divergence" configuration units (depicted by the thick black curves in Fig. 3 (d) and Fig. S3 (e)), which are not as clear as that in Fig. 3 (a). Nevertheless, corresponding vertical velocity could still perform strongly unified downward anomalies, partly suggesting that vertical velocity (ω) is a universal dynamical variable.


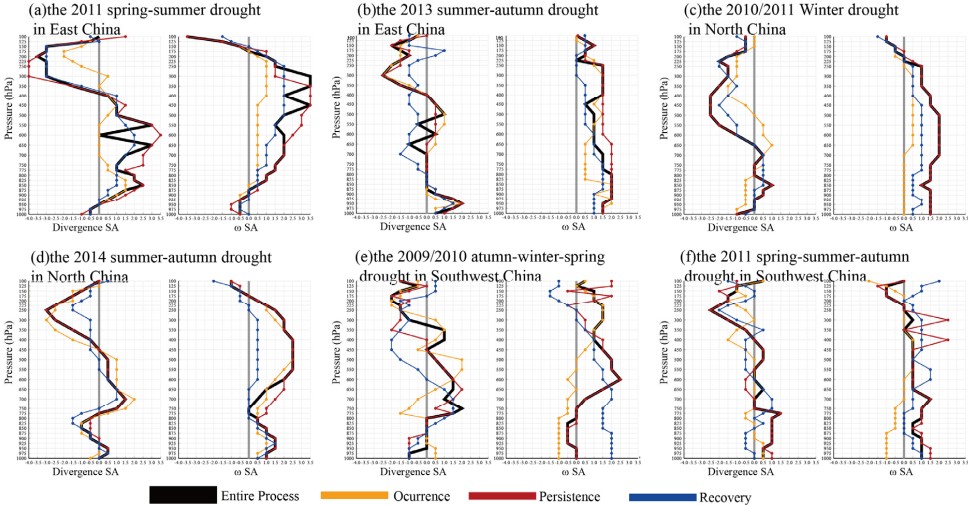

**Figure 3.** Vertical structures of Standardized Anomalies (SA) of season-scale horizontal divergence and vertical velocity (ω) regarding periods of drought occurrence, persistence, and recovery during six typical region-scale drought processes. The SA based values at each pressure level correspond to statistical modes of SA-based samples featuring (-4, 4] and 0.5 as the range and the interval. The number N
labelled in SA-axis actually means the range (N, N+0.5].

**4 Drought simulation using the SVVAI**

As commonly revealed in Sect. 3, vertical velocity (ω) has the potential of being a universal and simple drought indicator, and the SVVAI_max and SVVAI_ave are calculated to simulate droughts following the method in Sect. 2.5.





**4.1 Regional Processes**

Figure 4 and Fig. S4 showed good results of process simulation using the SVVAI, and even in some cases the SVVAI almost coincide with realistic drought processes(e.g., Fig. 4(b)), leading to relatively high correlation coefficients (Table 2). Unexpectedly, negative correlation coefficients occur during the 2013/2014 drought process in Xinjiang, indicating that the role of vertical motion in precipitation formation affected by geographical differences is not always positive.


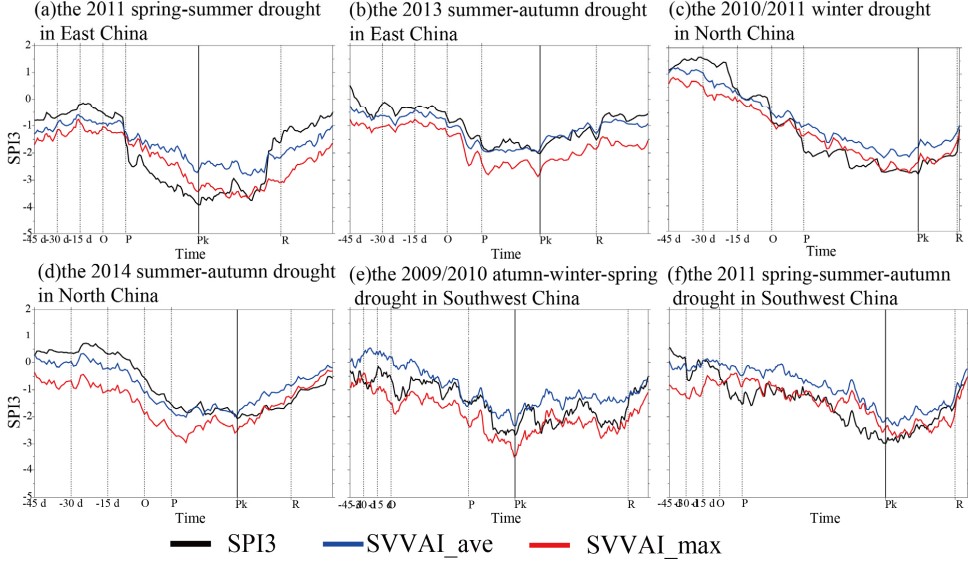

**Figure 4.** Season-scale process simulation using the SVVAI regarding six regional drought processes during 2009–2016 over China.

**Table 2.** Temporal correlation coefficients (TCC) of each regional process simulated with the SVVAI with respect to observed SPI3
during 2009–2016 over China

| Region | Names of Drought Processes | SVVAI_max | SVVAI_ave |
|---|---|---|---|
| East China | the 2011 spring-summer | 0.68 | 0.80 |
| | the 2013 summer-autumn | 0.90 | 0.90 |
| Inner Mongolia | the 2010 summer-autumn | 0.41 | 0.81 |
| North China | the 2010/2011 winter | 0.90 | 0.92 |
| | the 2014 summer-autumn | 0.69 | 0.75 |
| Northeast China | the 2011/2012 autumn-winter | 0.53 | 0.74 |
| Northwest | the 2013 spring | 0.68 | 0.80 |





| | | | |
|---|---|---|---|
| China | the 2015 summer-autumn | 0.47 | 0.67 |
| South China | the 2009 winter-spring | 0.79 | 0.82 |
| | the 2011 spring-summer-autumn | 0.65 | 0.80 |
| Southwest China | the 2009 winter-spring | 0.71 | 0.76 |
| | the 2009/2010 autumn-winter-spring | 0.85 | 0.82 |
| | the 2011 spring-summer-autumn | 0.82 | 0.90 |
| | the 2012 spring | 0.37 | 0.84 |
| | the 2012/2013 winter-spring | 0.25 | 0.71 |
| Tibet | the 2009 winter-spring | 0.58 | 0.61 |
| Xinjiang | the 2009 summer-autumn | 0.59 | 0.43 |
| | the 2013/2014 winter | -0.62 | -0.53 |

**4.2 Spatial distribution**

In addition to regional processes, performance on spatial distribution displayed in Figure 5 and Fig. S6 is another concern about the newly proposed drought indicator. Because grid-based simulation using the SVVAI_max lead to the systematically

dry tendency over China (Fig. S5), Fig. 5 and Fig. S6 only exhibit final results that spatially averaged values over China were removed from raw gridded simulation using the SVVAI_max.

As shown in Fig. 5 (a) and (b), the SVVAI_ave and SVVAI_max can spatially simulate Eastern-China-related droughts very well, indicating the quite correct intensities and locations. Regarding some certain droughts in Southwest China and Northeast China (Fig. 5 (e) and (f), and Fig. S6 (b)), the two schemes of the SVVAI both exhibit a little weaker intensities

but relatively correct locations, and the performance of SVVAI_ave is seemingly better than that of SVVAI_max. Also, there is incorrect indication of drought intensities and locations (e.g., the 2010/2011 North China drought), which cannot lead to the simple conclusion that the SVVAI failed to simulate spatial variations. The potential reason is that simulation performance is also time-varying, partly revealed by temporal evolutions of PCC coefficients (Fig. S7 and S8), and an example is the 2010/2011 North China drought simulated with the SVVAI_ave (Fig. S7 (c)), in which PCC reached +0.75 at

the drought occurrence but decreased to almost zero at the drought peak.



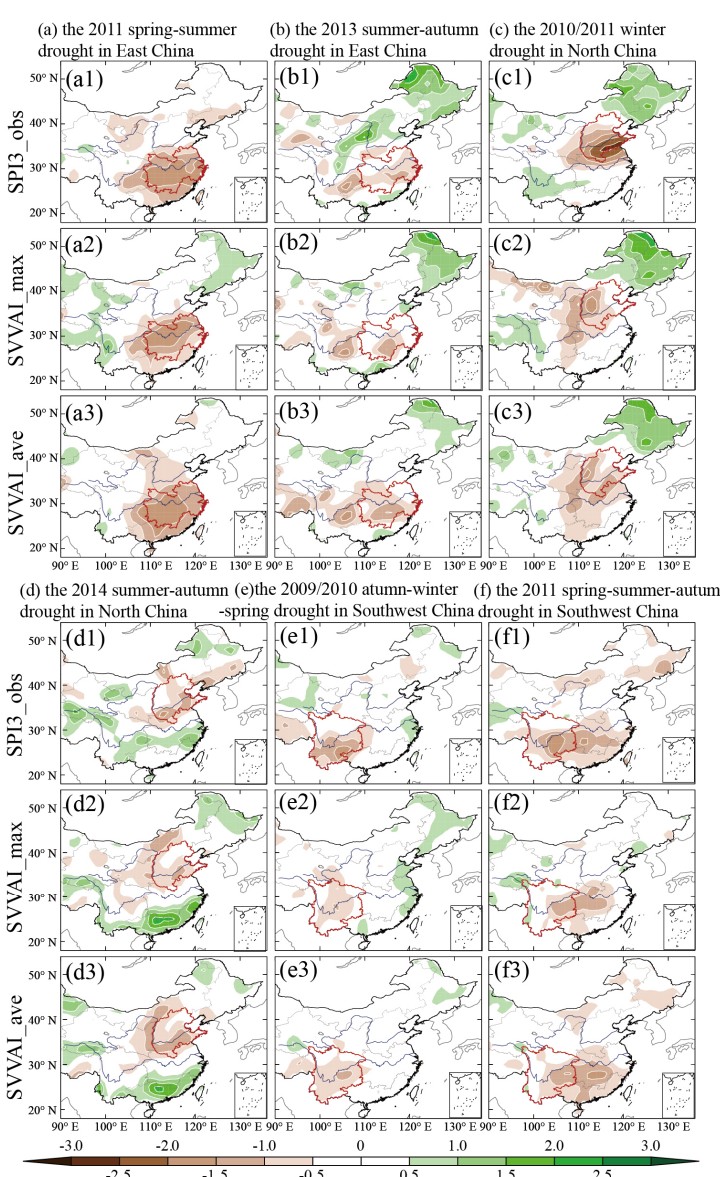



**Figure 5.** Composited spatial distribution simulated with the SVVAI regarding six regional drought processes during 2009–2016 over China. Target drought regions are marked with red thick curves. The results have been spatially smoothed with the nine-point smoother twice.

**5 Capacity of spatial simulation using the SVVAI and associated physical causes**

To further understand the performance of the SVVAI on spatial drought simulation in Sect. 4.2, TCC of the SVVAI against observed SPI3 at the grid scale were shown in Fig. 6. Season division herein is forty-five-day-decaying, when traditional seasonality and the fact that SPI3 reflects the past ninety-day precipitation anomalies are taken into account together. Main conclusions are as follow: (1) TCC generally decreases along the east–west direction (Fig. 6 (e) and (j)), and positive TCCs

above +0.3 occupied most areas to the east of 110º E, whereas large-range low TCCs (-0.1～+0.3) appear to the west of 110º E. No highly negative TCCs exist at the nation scale, partly indicating the positive contribution of vertical velocity ($\omega$) to precipitation formation. (2) It is notably seen that East China (i.e., the middle-low reach of Yangtze River basin) and Northeast China are regions where vertical velocity ($\omega$) show highly positive TCCs (+0.6～+0.8) with precipitation at the season scale, which can help understand results about drought simulation (e.g., good performance during the 2011 East

China drought (Fig. 5 (a))). Western pacific subtropical high (WPSH)(He et al., 2001) and northeast cold vortex (NECV)(Fang et al., 2018) are the large-scale circulation systems potentially responsible for these high TCCs. Physically, WPSH causes dynamical subsidence, while NECV corresponds to upward motion of converged airflow. Therefore, vertical motion herein is the important variable which help reveal the role of circulation systems in triggering or supressing precipitation formation. (3) The intensity of TCCs vary with time and space. TCCs temporally reach the local peak over

Tibet and Northeast China during 16 July～14 October, and large-area high TCCs (above +0.7) spatially appear over the southeastern part of China. (4) In addition, SVVAI_ave performed better than SVVAI_max to the east of 100º E, but both of them failed to highly correlate with SPI3 over the western part of China.



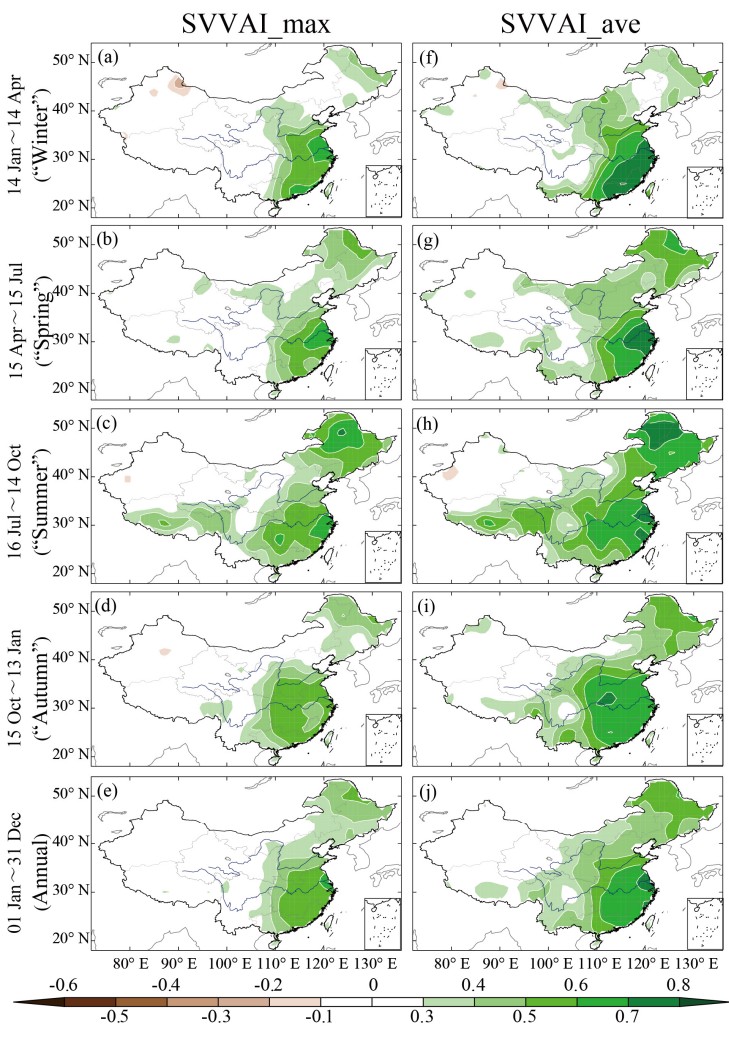

**Figure 6.** Spatial distribution of temporal correlation coefficients (TCC) of the SVVAI against observed SPI3 during 1981–2016 over China and associated seasonality. The results have been spatially smoothed with the nine-point smoother twice.





**6 Drought prediction using the average scheme of SVVAI (SVVAI_ave)**

As revealed in Sect. 4, the SVVAI could simulate regional drought processes and spatial distributions well, and the SVVAI_ave performed better than the SVVAI_max did. Further, drought prediction using the SVVAI_ave are preliminarily investigated herein, and the forecasted SPI3, a traditional precipitation-based drought indicator calculated on the basis of CFSv2 forecasted precipitation, is also employed for comparison.

**6.1 Regional Processes**

Figure 7 and Fig. S9 show results of the prospective 60-day process prediction, in most of which performance of the predicted SVVAI_ave seems to be equally matched with or outperformed that of the forecasted SPI3. Further, quantitative assessment was implemented (Fig. 8 and Fig. S10), when mean and standard deviation (SD) of sample bias are considered.

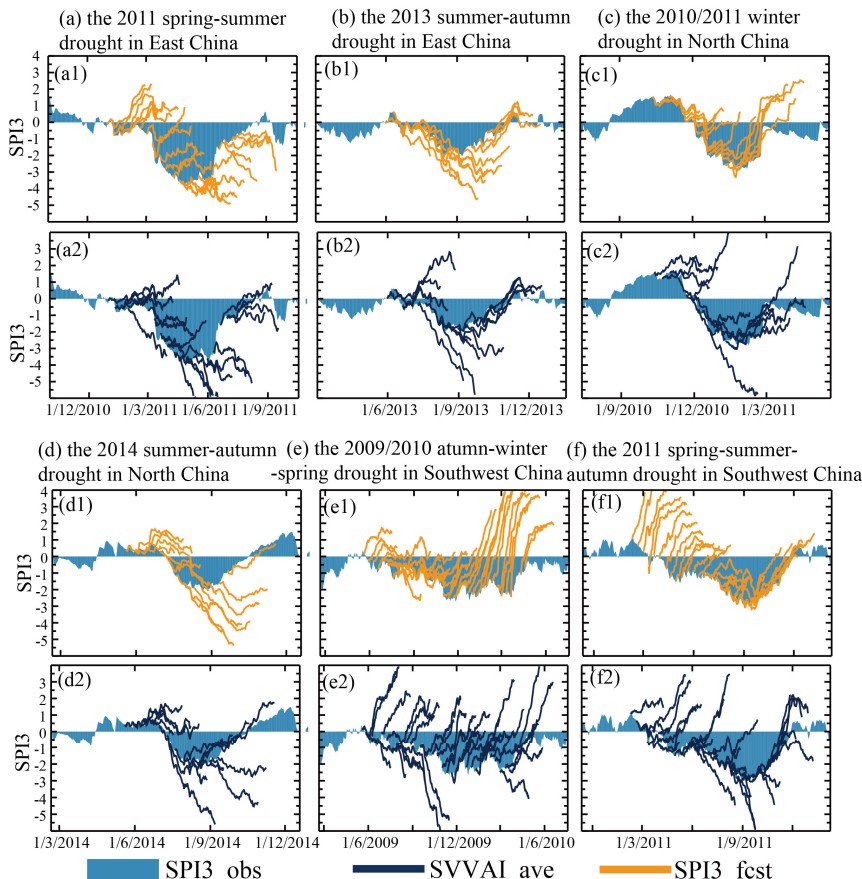

**Figure 7.** The prospective 60-day prediction of season-scale processes using the SVVAI_ave regarding six regional drought processes during 2009–2016 over China.

Mean values of sample bias from the forecasted SPI3, compared with those of the predicted SVVAI_ave, displayed a relatively dramatic increase over Southwest China (Fig. 8 (e) and (f), and Fig. S10 (g), (h), (i)) and Northwest China (Fig. S10 (c)), easily explained by systematic bias from CFSv2 precipitation products over the climatologically dry regions (e.g., Northwest China) and topographically complex regions (e.g., Southwest China). Additionally, the growth rates of sample bias from predicted SVVAI_ave over East China and North China (Fig. 8 (b), (c) and (d)) are slower and steadier than those of the forecasted SPI3. The predicted SVVAI_ave might therefore be a better drought predictor in this situation.





It is interesting that some characteristics of the predicted SVVAI_ave on the basis of standard deviation (SD) of sample bias

were seemingly common. In almost all the cases, SD curves of the forecasted SPI3 and predicted SVVAI_ave (dashed ones in Fig. 8 and Fig. S10) obviously cross before the 60th day. Before the cross-points depicted by grey dots, sample bias of the predicted SVVAI_ave exhibited less SD values, with time-varying critical times ranging from 15th day to 55th day.

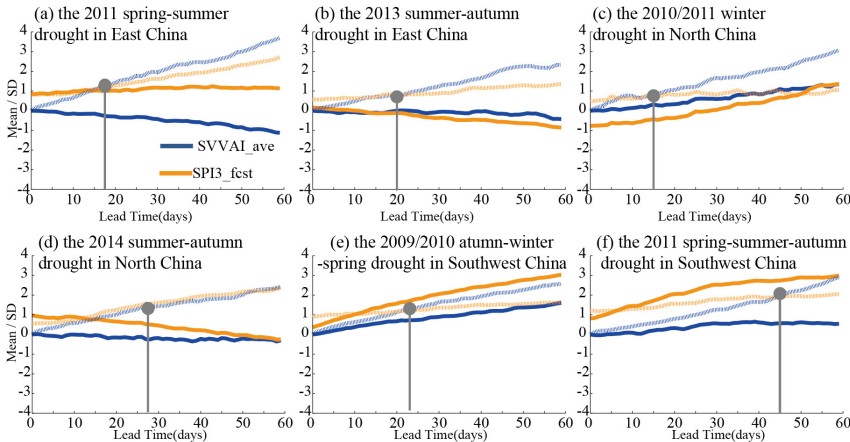

**Figure 8.** Temporally lead-time based evolution of mean (solid curves) and standard deviation (SD; dashed curves) of prediction minus observation bias. For one point of each curve, mean and SD are calculated based on all samples bias at corresponding lead times. Grey dots indicate cross points of the two SD curves.

## 6.2 Spatial Distributions

The 2011 summer-autumn drought in Southwest China are herein employed for forecasting assessment of spatial distribution, which is a good case that the 2011 spring-summer drought in East China and considerably large-scale drought signals over Northeast China and Inner Mongolia could also be observed. Prediction performance at the occurrence, peak and termination times exhibit in Fig. 9~Fig. 11.

At the occurrence time, the forecasted SPI3 indicated extreme wetness but no drought signals in the northern part inside

Southwest China, extending to Northwest China. In comparison, performance of the predicted SVVAI_ave approached observation, suggesting slightly scattered wetness or dryness within Southwest China. At the same time, the 2011 spring-summer drought over East China still persist and large-range extreme drought signals cover the entire region, the realistic intensity and coverage of which can seemingly be predicted by the SVVAI_ave from the preceding 29th to the 9th day.

When it comes to the peak time (Fig. 10), the severe drought centre located over the southeastern corner of Southwest China,

and anomalously dry signals extended to almost the entire region. Both the forecasted SPI3 and predicted SVVAI_ave could





indicate the severely dry centre from the 26<sup>th</sup> to 6<sup>th</sup> day before, but the predicted SVVAI_ave performed a little worse mainly due to overestimation of drought intensity. At the termination time (Fig. 11), the forecasted SPI3 can also indicate severe drought signals over Northeast China, and the forecasted intensity and coverage were correct and steady even with longer lead times.



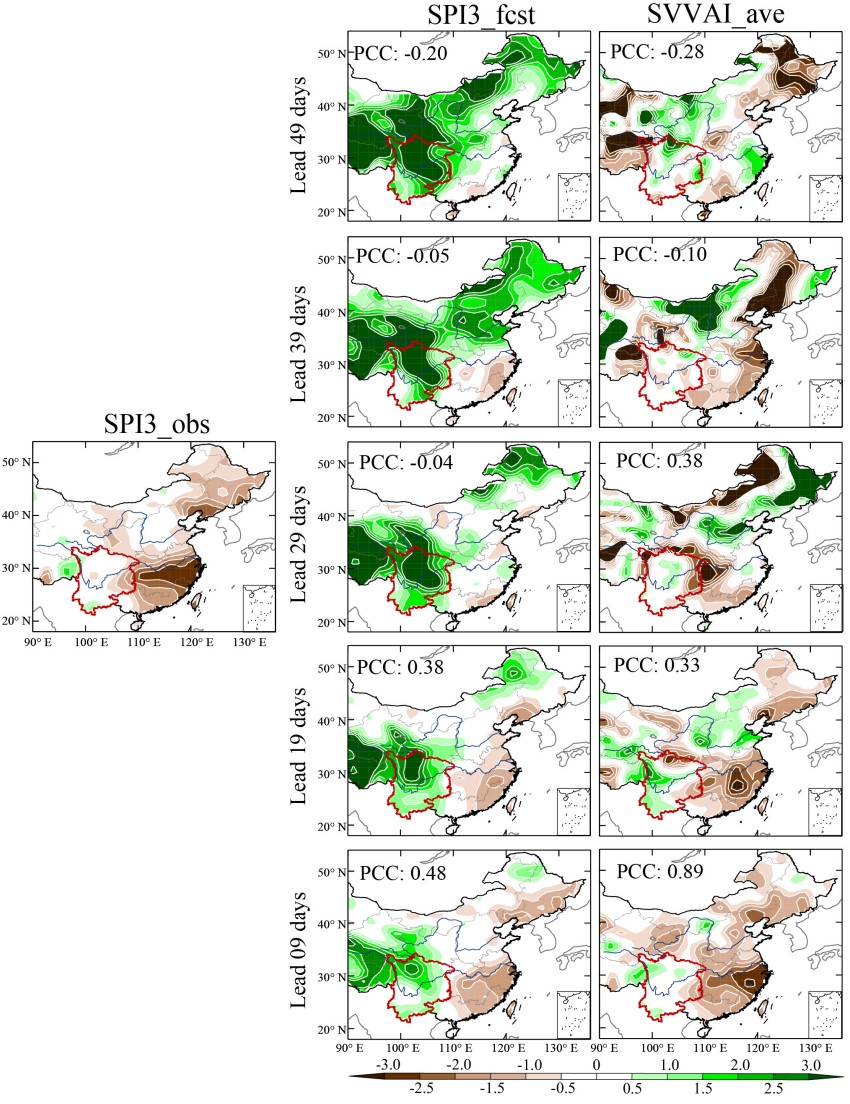

**Figure 9.** Spatial prediction at the occurrence time of the 2011 summer-autumn drought over Southwest China using the predicted SVVAI_ave and the forecasted SPI3 with different lead times. Pattern correlation coefficients (PCC) are computed on the basis of gridded values within Southwest China (thick red curves). The results have been spatially smoothed with the nine-point smoother twice.






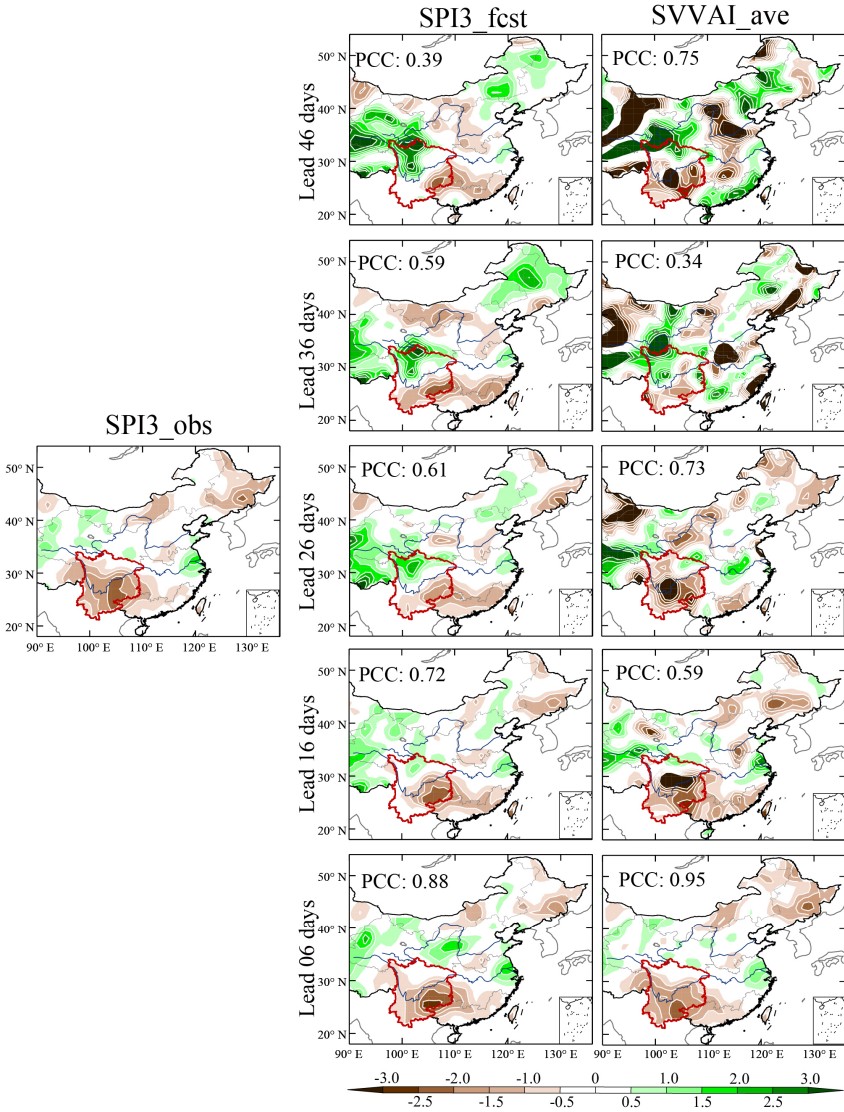

**Figure 10.** Same as Figure 9, but for prediction at the peak time of the 2011 summer-autumn drought over Southwest China.

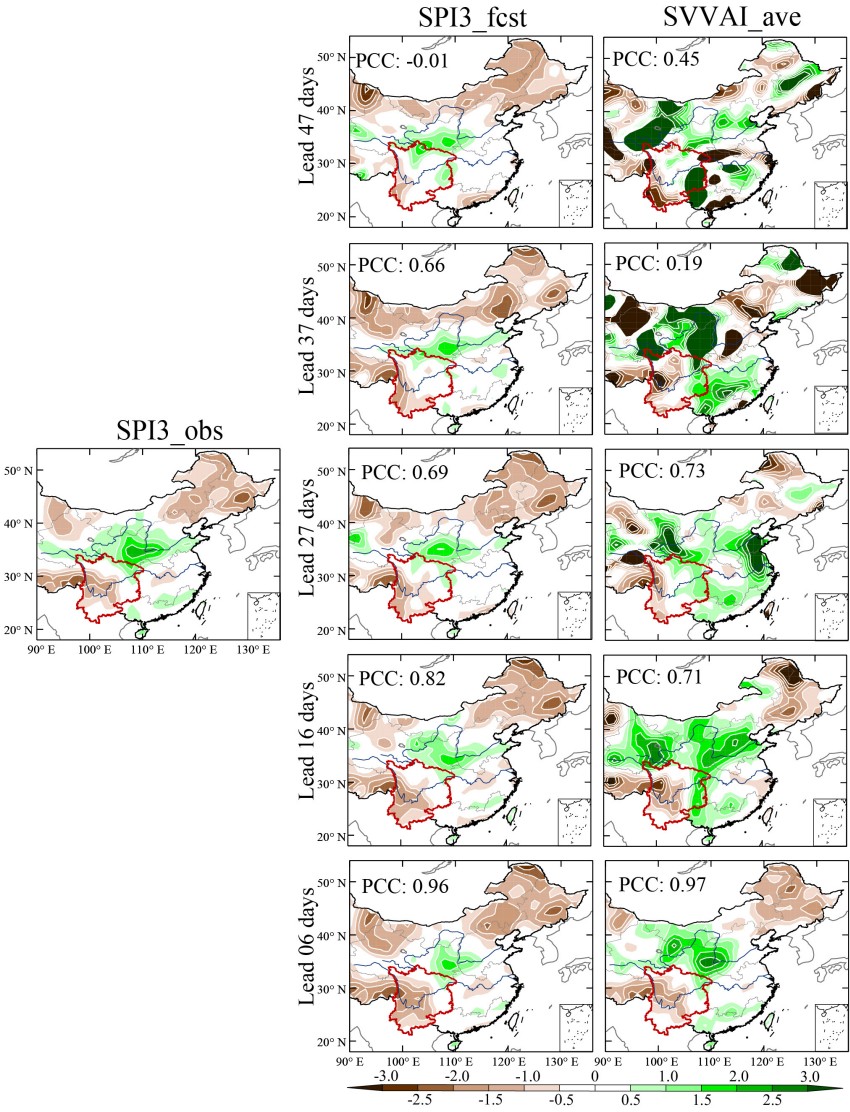

**Figure 11.** Same as Figure 9, but for prediction at the termination time of the 2011 summer-autumn drought over Southwest China.



## 7 Discussion

Considering that persistent dynamical subsidence of air current is physically related to long-term precipitation deficit,
vertical velocity (ω) is employed to explore issues associated with meteorological droughts in the present study. Spatio-
temporal evolution of dynamical anomalies were uncovered, which help further understand the role of atmospheric dynamics
in drought development. Furthermore, drought simulation and prediction were preliminarily performed using the novel
Standardized Vertical Velocity Anomaly Index (SVVAI), having the potential of being a drought indicator. However, the
aforementioned methods and achievements are a new attempt with many shortcomings, and some relevant issues to further
discuss and understand are as follow:

The first topic is physical meanings of indicators. So far, many drought indices have been proposed, some of which are
physically based. For example, EDDI (Evaporative Demand Drought Index) (McEvoy et al., 2016;Hobbins et al., 2016)
proposed recently is such an index measuring drought signal through the response of atmospheric evaporative demand $E_0$ to
surface drying anomalies. Similarly, the SVVAI newly proposed herein links persistent atmospheric dynamical subsidence to
long-term precipitation deficit. Since hydro-climate processes that influent formation and development of meteorological
drought are multiple and complex, both the EDDI and SVVAI are constructed based on one of physically meaningful aspects,
which are overall useful in drought monitoring and prediction as expected.

Second, the quantitative relationship between ω anomalies and drought indices could be improved via parameter calibration
and selection of pressure levels. (1) The proposed SVVAI could further be calculated on the basis of different computation
schemes. The vertically average scheme herein made SVVAI_ave a simple drought indicator, which did not always
performed very well. Further efforts could be made to calibrate non-linear and comprehensive relationships between ω
anomalies and drought indices, with the help of machine learning-related methods. In addition, parameter weights retrieved
from the calibrated relationship can also help discover which typical pressure levels where vertical motion affect
precipitation formation a lot. (2) Selection of pressure levels is a non-negligible detail. Vertical intervals of pressure levels
for numerical computation is not equal (Table 1), leading to the fact that weights of ω anomalies in the upper and lower
pressure levels were strengthened in nature when the average scheme chose. In addition, the lower limitation designed to be
the minimum pressure value between 850 hPa and surface pressure (Sect. 2.5), together with spatially heterogeneous
distributions of surface pressure levels (Fig.1 (b)), might lead to bad performance on drought simulation. For example, weak
intensities and wrong spatial simulation over the 2010/2011 North China drought (Fig. 5 (c)) could be partly explained by
the fact that ω anomalies below 850 hPa were not considered.

The third issue to explore is index applicability. Since SPI is a conventional and powerful index widely used in drought
monitoring and prediction which cannot simply be replaced with the concurrent SVVAI, our concern is to explore when and
where the predicted SVVAI outperformed the forecasted SPI3 while the forecasted SPI3 does not work, such as more
realistic prediction of spatial distribution using the SVVAI during the 2011 Southwest China drought (Fig. 9). The SVVAI is


worthy of being deeply investigated to find similarly exceptional cases, suggesting its complementary contributions to current studies about drought monitoring and prediction.

Fourth, research significance of the present study is to provide an insight for antecedent signals of potential drought prediction. Analysis on concurrent dynamical anomalies can not only reveal its potential of being a natural precursor, but also contribute to the discovery of antecedent drought-inducing atmospheric/oceanic anomalies. (1) Dynamical anomalies and SVVAI concurrent with drought development have no lead times, forced with the forecasted $\omega$ products for drought prediction. Even so, dynamical anomalies still can be a precursor, since it is notable that dynamically anomalous configurations (e.g., anomalies of horizontal divergence) during drought processes had appeared and lasted for several days before drought occurrence ((a), (d), and (e) of Fig. 2). (2) Vertical motion, the vertical branch of realistic three-dimensional atmospheric circulations, is positively related to drought development (Fig. 2 and Table 2), which help understand the role of atmospheric circulations in drought development. Specifically speaking, in order to further explore antecedent drought-inducing atmospheric/oceanic anomalies, seasonal evolution of atmospheric circulations (e.g., the typical Hadley and Walker circulations) can be focused, which may provide a fundamental approach for drought prediction with long lead times.

Fifth, analysis on evolution of atmospheric dynamical anomalies herein (Fig. 2) may contribute to the investigation on the role of land-surface–atmosphere interaction in drought development at the seasonal scale with a daily running window. One assumption previously mentioned in Long et al. (2000) suggest that changes in the general circulations initiate drought, while other mechanisms (e.g., land-atmosphere interactions) contribute to drought persistence. Similarly, when Zeng et al. (2019) studied effects of land–atmosphere coupling on northeastern China persistent drought in spring–summer of 2017, it suggest that an upstream quasi-stationary wave pattern strengthened by land–atmosphere coupling should be considered in diagnosing persistent droughts over northern mid-latitudes (Zeng et al., 2019a). When it comes to Fig. 2 herein, anomalous signals of horizontal divergence and vertical motion at drought occurrence were indeed maintained and strengthened at drought persistence, indirectly providing feasible examples for investigation on effects of land-surface–atmosphere interactions in drought development.

Sixth, the application of the SVVAI in current operational drought prediction need to be improved. It is necessary to further verify and correct the CFSv2 forecasted $\omega$ sub datasets via post processing techniques and ensemble forecast (see Sect. 5.1.2 and 5.1.3 in Hao et al. (2018)). It might help improve the performance of the predicted SVVAI a lot, since results herein were on the basis of raw forecast products.

## 8 Conclusion

Anomalous vertical motion is one dynamically direct drought-inducing cause, which has the potential of being a universal drought indicator. To investigate it, we chose severe droughts during 2009–2016 over all the drought study regions of China as the research targets. Three-month SPI (SPI3) updated daily was used to identify severe regional drought processes at the



season scale, and accordingly the original values of vertical velocity (ω) and associated horizontal divergence were also transformed to season-scale standardized anomalies (SA) with a daily running window. Subsequently, the present study tried to explore the two key issues: one is to analyse dynamically drought-related vertical structures and associated temporal

evolutions, and the other is to simulate and preliminarily predict droughts using vertical motion-based indices. In particular, drought processes and spatial patterns are our concerns. To date, the main results and conclusions are as follow:

Firstly, vertical motion and associated horizontal divergence were employed to investigate atmospheric dynamical anomalies during region-scale drought processes and key phases. (1) Dynamically vertical structures are characterized as the typically anomalous "upper-convergence–lower-divergence" pattern and the intensified downwards vertical motion as expected. (2)

Signal intensities and relative positions of dynamically vertical structures are time-varying, the temporal evolutions of which mostly coincide with regional drought processes. Anomalous intensities of both vertical velocity and horizontal divergence reach almost maximum values usually near the time of drought peak but are relatively weak at drought occurrence or recovery. In particular, when evolved from persistence to recovery, the SA-based vertical structure of horizontal divergence and the positive SA of vertical motion get weakened, rather than the pattern-reversed divergence structure or obviously

negative ω SA. (3) Comprehensively speaking, SA-based vertical structures of horizontal divergence during drought processes exhibit one or more "upper-convergence–lower-divergence" configuration units. Nevertheless, corresponding SA of vertical velocity could still perform strongly unified downwards anomalies, partly suggesting that vertical velocity is a potentially universal dynamical variable.

Secondly, the SVVAI (Standardized Vertical Velocity Anomaly Index) was newly proposed to simulate and predict droughts,

based on newly uncovered knowledge of dynamically drought-inducing mechanism. The SVVAI is calculated using SA-based values of vertical motion at multiple pressure levels in the troposphere. With respect to specified computation schemes, the SVVAI_ave and SVVAI_max, corresponding to the vertically average- and maximum-based methods, were adopted for drought simulation.

The third research work is to simulate processes and spatial distributions of all the regional droughts over China using the

SVVAI_ave and SVVAI_max. (1) Drought processes simulated by both SVVAI_max and SVVAI_ave show highly positive correlation coefficients with realistic droughts over most regions, and the SVVAI_ave outperformed the SVVAI_max. (2) Simulation performance of composited spatial distributions is seemingly related to geographical difference. They could indicate quite correct intensities and locations regarding East-China-related droughts, and they exhibit a little weaker intensities but relatively correct locations over some certain droughts in Southwest China and Northeast China. However,

there is incorrect indication of drought intensities and locations (e.g., the 2010/2011 North China drought).

Fourthly, to help understand the performance of the SVVAI on drought simulation, temporal correlation coefficients (TCC) of the SVVAI against observed SPI3 at the grid scale were computed. (1) Positive TCCs above +0.3 occupies most areas to the east of 110º E, while large-range low TCCs (-0.1～+0.3) appear to the west of 110º E. With respect to simulate capacity, SVVAI_ave performs better than SVVAI_max to the east of 100º E, but both of them fail to correlate with SPI3 over the

western part of China. (2) It is notably seen that East China (i.e., the middle-low reach of Yangtze River basin) and



Northeast China are the two regions with highly positive TCCs (+0.6~+0.8). Western pacific subtropical high (WPSH) and northeast cold vortex (NECV) are the large-scale circulation systems potentially responsible for these high TCCs. Physically, WPSH causes dynamical subsidence, while NECV corresponds to upward motion of converged airflow.

Fifthly, the application of the SVVAI_ave in drought prediction was preliminarily explored. The SVVAI itself has no lead times but depends on that of climate forecast products, which means that the predicted SVVAI_ave could be calculated based on vertical velocity retrieved from a combination of the CFSv2 forecast products and ERA-Interim reanalysis datasets. In addition, the forecasted SPI3 is also employed for comparison, calculated based on a combination of the CFSv2 forecast products and real-time CMA observation. Primary achievements are as follow: (1) With respect to the prospective 60-day prediction of regional processes, performance of the predicted SVVAI_ave seems to be equally matched with or outperform that of the forecasted SPI3. (2) Prediction of spatial distribution is preliminarily assessed via the example of the 2011 summer-autumn drought over Southwest China. Performance of prediction at the occurrence, peak and termination times are inconsistent. The SVVAI_ave predicted almost realistic drought intensities and coverage than the forecasted SPI3 did at drought occurrence, whereas it performed a little worse than the forecasted SPI3 mainly due to overestimation of drought intensities at drought peak and shorter lead time of predicting drought termination.

Overall, the newly proposed SVVAI herein has the potential of being a universal drought indicator, complementary to current approaches of operational drought monitoring and prediction. Further study could be focused on the following two aspects at least: One is index applicability, that is to say, to explore when and where the predicted SVVAI outperform the forecasted SPI3. The other is to further explore antecedent drought-inducing signals of atmospheric/oceanic anomalies with the bridge of vertical motion, which may provide a fundamental approach for drought prediction with long lead times.

*Data availability.* All the datasets used in this paper are publicly accessible.

*Author contributions.* ZL, with the help of HH and ZW, designed the research. HH and ZW obtained the funding supporting it. ZL conducted case analysis and wrote the original draft. HH, ZW, and GL jointly reviewed the entire manuscript. HY provided ideas for model improvement and help write the discussion part. All authors finally read and approved the paper.

*Competing interests.* The authors declare no competing interests.

*Acknowledgements.* This work is supported by the National Key R&D Program of China (grants 2017YFC1502403 and 2018YFC0407701), the National Natural Science Foundation of China (grants 51579065 and 51779071), and the Water Conservancy Program of Jiangsu Province (grants 2015019 and 2017007).





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
