# Peer review of "The Standardized Vertical Velocity Anomaly Index (SVVAI): Using Atmospheric Dynamical Anomalies to Simulate and Predict Meteorological Droughts"

_Earth System Dynamics, 2019_

## Referee Comment (RC1) · Paolo De Luca (Referee) · 24 Jan 2020

**Review of ESDD manuscript: "The Standardized Vertical Velocity Anomaly Index (SVVAI): Using Atmospheric Dynamical Anomalies to Simulate and Predict Meteorological Droughts" by Liu et al.**

The authors propose a new drought index (SVVAI) derived from vertical velocity anomalies and compare it with the 3-month Standardized Precipitation Index (SPI-3). Although the overall idea of the SVVAI may be original and the quantified performance of the newly introduced index seems to be fairly good, the manuscript has major issues that, in my opinion, make it not suitable for publication in ESD.

Please find below my major concerns:

(1) The English is poor, so that while reading the paper I had difficulties in understanding many sentences and their key messages. Punctuation sometimes is incorrect and references are not well incorporated within the text. I suggest to let a native English-speaker check carefully the whole text before a possible re-submission.

(2) At the end of the manuscript (Discussion/Conclusion(s) sections), it is not clear to me what really are the advantages that the SVVAI may bring to for example operational forecast of drought, compared to the SPI-3. Moreover, the Discussion section only highlights the limitations of the SVVAI. It is good to mention them, but I would have expected at least a balance between pros and cons.

(3) The SVVAI has been compared to the SPI-3, but how it performs compared to the other SPI indices (e.g. SPI-1 and SPI-6)? And how it performs more generally compared to other indices of drought, such as the PDSI and SPEI?

(4) What the authors can say about the fact that SVVAI is computed with daily observations, whereas the SPI is computed with monthly data, by also including a time-lag in this case of 3 months?

(5) In Section 2.5, the main definition of the SVVAI index is not clear. Equations 2-3 need to be amended to fully reflect the SVVAI definition. What is the range of values of the SVVAI? Is it the same as the SPI or different? This is again not clear and very important, because many figures show SVVAI and SPI-3 values on the same range of values and colors.

(6) Within the analyses the authors made use of the Temporal Correlation Coefficient (TCC) and Pattern Correlation Coefficient (PCC), but no key references are provided with respect to these two statistical tests. In addition, statistical significance (p-values) of these correlations are not provided. Therefore, it is difficult to quantify the robustness of the results.

(7) The Figure captions are not exhaustive, so that it is difficult to interpret the plots. I suggest adding more information so that the reader can understand the plots without the need to refer to other sections of the paper.

---

## Referee Comment (RC2) · Anonymous Referee #2 · 15 Jun 2020

This paper introduces a new index (SVVAI) that quantifies the tropospheric vertical velocity, with the application of this index as a meteorological drought diagnostic. The SVVAI is compared to the 3-month Standardised Precipitation Index (SPI3) for six case-study drought events in China. The potential for the SVVAI in drought prediction is also explored for these drought events. The SVVAI is a novel approach to diagnosing drought, with most drought indices based on surface variables rather than atmospheric. Unfortunately, I found the execution of the article to be poor, and recommend rejection.

Major comments

1. The English is not of the required standard. It is frequently a significant challenge to interpret the text, requiring very careful reading. This makes it very difficult to focus on the intended presentation and interpretation of results. I suggest the authors improve the grammar before resubmission.

2. The conclusions for both drought simulation and prediction are hampered by sample size. By selecting only a few case studies, it is difficult to draw overall conclusions about the relative benefits of SPI3 versus SVVAI. For example, how do the two indices compare during (a) other droughts, and (b) on all days (not just droughts)?

Furthermore, the authors' conclusions about the SVVAI being equal or superior to the SPI3 in predictive performance (L316) to be overstated. With only a small sample of events and no use of forecast verification metrics, it is impossible to draw this kind of conclusion.

3. I do not think the authors adequately describe the reasoning behind using the horizontal divergence and vertical velocity as diagnostics of drought. Dynamically, how do we expect these variables to change during drought? Is that seen in the results presented here (e.g. Fig 2). The paper could do more to link the results to dynamical processes. For me this is a "major comment" because it underpins the whole point of the paper.

4. There should be greater discussion of the relative benefits of SVVAI and SPI3. Why would you use SVVAI when you could just use SPI? In what circumstances is it preferable to use SVVAI? Can you draw those conclusions from the results presented here?

For example, Fig 6 implies that SVVAI is only comparable to SPI3 for eastern China, but there is no discussion of this. Does this mean SVVAI is not a useful indicator of meteorological drought for the rest of China?

5. There is too much information presented across all the figures. I think some of the

information should be synthesised. For example, is it really necessary to have 3 similar figures, each with 11 panels (Figs 9, 10, 11)? Surely there are ways of presenting this information more concisely. I also do not advocate simply moving some figures to the supporting information, as that already contains many results.

6. I find the conclusions of the forecasting section to not match the results. As mentioned in point 2, the statements about predictive performance are severely hampered by the sample size of events and lack of forecast verification metrics.

Greater synthesis of the forecast results is needed. For example, for Figs 9, 10 and 11, the authors draw conclusions about model performance by visual inspection of spatial fields. While it is useful to visualise it this way, I do not think it needs to be done 3 times over (this is related to point 5). In fact, the authors already use a measure of pattern correlation (PCC). Could PCC not be used to quantify the similarity of forecast fields to the observation, or between the two models' forecast fields, as a way of presenting results more generally?

Furthermore, I am concerned by the lack of forecast calibration. To be fair, the authors note that the CFSv2 forecasts used in SVVAI prediction are raw (L423-426), and could benefit from calibrating to the ERA-Interim vertical velocity and horizontal divergence data. However, it is not mentioned that the precipitation forecasts are also raw, and hence the SPI3 also require post-processing to indicate operational forecast skill. It is entirely possible (probable, even?) that the biases in the precipitation forecasts are larger than the biases in the atmospheric variables (in fact I think this is what the authors are saying in L83-85). Therefore, the forecast results are presented with the models on an unequal footing, making comparisons very difficult.

Minor comments

1. Figure captions are not detailed enough. For example, what is Figure 7 showing? Why the many different orange and black lines? I assume they are different initialisation dates but please describe what's shown in the figures.

2. L153: What does the "critical proportion" mean? Please explain. Is this where you define "extreme" and "severe" drought processes?

3. Figure 2 caption: "shadow" should be "shading".

4. Figure 4, 7: The y-axis label should surely be something like "Standardized anomaly" rather than SPI3?

5. Section 6: Remind readers which forecast model is used (CFSv2) at the beginning of the section.

6. L316: How do the authors come to the conclusion that SVVAI_ave is equal or superior to SPI3? Please walk the reader through the figure so that they can draw the same conclusions.

---

## Referee Comment (RC3) · Anonymous Referee #3 · 5 Jul 2020

This manuscript discusses the dynamics of droughts over different climate regimes in China. The authors depart from the many typical drought studies that are based on surface indices, offering a new and refreshing approach based on atmospheric dynamical principles. They propose to use vertical velocity and convergence/divergence patterns to define two new drought indices. I find the concept exciting and promising.

The study uses global reanalysis data to assess the general behavior of the proposed index against the traditional approach of using SPI. Then, they employ long-term operational forecasts to evaluate whether the new indices contain a predictive signal, in

which case they could be used as a drought forecast tool.

The analysis is carried over several case studies and thus cannot offer a statistical significance. Some regions of China towards the east seem to respond consistently to the dynamical hypothesis, but this is not the case of droughts in other areas towards the west. The authors are honest and cautious about the possible limitations of the approach. In this reviewer's view, the study is a promising approach that has the potential to complement traditional drought studies. In that sense, I would not take the results as a definitive answer but as the starting point for other studies in this matter.

I would argue that the manuscript meets most of the review criteria for this journal, as defined in https://www.earth-system-dynamics.net/peer_review/review_criteria.html, with one exception. The manuscript is, in the most part, understandable despite limitations with the language and grammar. My suggestion to the authors is to get help from an editorial office or native English speaker that can review and help correct the grammar. They could also use software like Grammarly that helps detect and offer suggestions to many of the weakly formed sentences. Grammarly, and likely other equivalent software, offer subscribers additional support from an expert team at a fee.

My recommendation is that the manuscript should be published after those corrections. It would be a loss if poor grammar were used as the main factor to prevent publishing.

---

## Author Comment (AC1) · 20 Jul 2020

**Response to comments from Referee#1:**

The authors propose a new drought index (SVVAI) derived from vertical velocity anomalies and compare it with the 3-month Standardized Precipitation Index (SPI-3). Although the overall idea of the SVVAI may be original and the quantified performance of the newly introduced index seems to be fairly good, the manuscript has major issues that, in my opinion, make it not suitable for publication in ESD.

Response:

Thank you for the comments. The paper proposed a new drought index and preliminarily investigate its performance in case studies. The approach is not perfect but we try to display some discoveries and the potential in drought prediction. With the help of your detailed major concerns, we will solve the issues listed below and improve the manuscript.

Please find below my major concerns:

(1) The English is poor, so that while reading the paper I had difficulties in understanding many sentences and their key messages. Punctuation sometimes is incorrect, and references are not well incorporated within the text. I suggest to let a native English-speaker check carefully the whole text before a possible re-submission.

Response:

We surely check it firstly with the software called Grammarly, and then surely seek help from native English-speaker to make it readable.

(2) At the end of the manuscript (Discussion/Conclusion(s) sections), it is not clear to me what really are the advantages that the SVVAI may bring to for example operational forecast of drought, compared to the SPI-3. Moreover, the Discussion section only highlights the limitations of the SVVAI. It is good to mention them, but I would have expected at least a balance between pros and cons.

Response:

The referee#1 's concern is the advantages the SVVAI brings to the operational drought forecast. Case studies in section 6.1 displayed preliminary results. That is, the forecasted region-scale SVVAI performs more steady (less standard deviation and less bias) within limited lead times when compared with SPI. Also, the SVVAI perform the potential in the forecast spatial distributions of the 2011 megadrought over southern China in section 6.2. Cases studies at the region and grid scales tend to display the potential of application of the concurrent SVVAI in forecasting.

Indeed, this comment helps us realize that more explicit conclusions are needed, and we will further make it clear and brief.

(3) The SVVAI has been compared to the SPI-3, but how it performs compared to the other SPI indices (e.g. SPI-1 and SPI-6)? And how it performs more generally compared to other indices of drought, such as the PDSI and SPEI?

Response:

SVVAI employed in case studies are three-month, consistent with the same timescales of SPI3. That is, SVVAI and SPI in the paper are always concurrent and have the same timescales. We will surely make it clear in the manuscript to avoid possible misunderstandings. Thank you for the comment.

The physically explicit and comprehensive PDSI and SPEI are indeed useful and popular in drought communities. However, surface air temperature is a fundamental part when computing these two drought indices via the intermediate variable of potential evapotranspiration. Our original idea is that we did not choose PDSI and SPEI as the target index because we try to avoid the consideration of surface air temperature when proposing SVVAI. Specifically speaking, the relationship between atmospheric dynamical subsidence and precipitation deficit is physically explicit, but it might be not always the case for the influence of dynamical subsidence on surface air temperature. Indeed, the considerations of PDSI and SPEI can help

understand the mechanisms behind hot droughts (i.e., concurrent drought and heatwave) or issues related to climate change, and we will illustration it in the discussion part.

(4)  What the authors can say about the fact that SVVAI is computed with daily observations, whereas the SPI is computed with monthly data, by also including a time-lag in this case of 3 months?

Response:

Both of SVVAI and SPI employed herein are three-month (90-day in practice) scale updated daily. That is, the index located on 1$^{st}$ April 1999 is calculated originally based on the 90-day values from 2$^{nd}$ Jan 1999 to 1$^{st}$ April 1999.

This comment helps us realize that section 2.3 may be confusing and not readable. We will modify it and give a clear and brief description.

(5)  In Section 2.5, the main definition of the SVVAI index is not clear. Equations 2-3 need to be amended to fully reflect the SVVAI definition. What is the range of values of the SVVAI? Is it the same as the SPI or different? This is again not clear and very important, because many figures show SVVAI and SPI-3 values on the same range of values and colors.

Response:

We understand your concern about the definition in Eq. 2-3. In particular, sub-indices herein is implicit, and we will list all of them in the next version.

We did not investigate the value ranges of the SVVAI and difference compared with the SPI. Since performance in Figure 4-5 and Figure S4-5 can indirectly indicate good index applicability, we did not consider it. Even so, we think the comment is constructive and help make the SVVAI more strict. We will claim it in the discussion part and improve it in further study. Thank you for this comment.

(6)  Within the analyses the authors made use of the Temporal Correlation Coefficient (TCC) and Pattern Correlation Coefficient (PCC), but no key references are provided with respect to these two statistical tests. In addition, statistical significance (p-values) of these correlations are not provided. Therefore, it is difficult to quantify the robustness of the results.

Response:

Key references and relevant details are provided in the supplement file, as illustrated in LINE 211. Therein we tell readers can find details about PCC (i.e., Anomaly Correlation) in section 8.6.4 of the book (Wilks, 2011). Anyway, we will move the key reference from supplement files to section 2.6 to make it clear.

Also, the statistical significance of correlations will be provided.

References:

Wilks, D. S. (2011). *Statistical Methods in the Atmospheric Sciences* (3 ed. Vol. 100): Academic Press.

(7)  The Figure captions are not exhaustive, so that it is difficult to interpret the plots. I suggest adding more information so that the reader can understand the plots without the need to refer to other sections of the paper.

Response:

Thank you for the comment, and we will further make the figure captions as exhaustive as possible.

---

## Author Comment (AC2) · 20 Jul 2020

**Response to comments from Referee#2:**

This paper introduces a new index (SVVAI) that quantifies the tropospheric vertical velocity, with the application of this index as a meteorological drought diagnostic. The SVVAI is compared to the 3-month Standardised Precipitation Index (SPI3) for six case-study drought events in China. The potential for the SVVAI in drought prediction is also explored for these drought events. The SVVAI is a novel approach to diagnosing drought, with most drought indices based on surface variables rather than atmospheric. Unfortunately, I found the execution of the article to be poor, and recommend rejection.

Response:

Thank you for highlighting the novelty of this paper. It is a new attempt but not perfect indeed.

Regarding the poor execution of the article mentioned by Referee#2, further studies have been conducted and some preliminary results have been achieved. However, it is another specified research that I can say too much herein.

Most of the detailed comments are constructive, and we appreciate it. Some of them surely facilitate the further modifications in the next version, while those issues that we are unable to solve in one single paper will be displayed in the discussion part.

Besides, some comments indicate that some misunderstandings happen mainly due to different academic backgrounds (e.g., concerns about the roles of dynamical climate diagnose in Figure 2 and Figure S2). Also, relevant illustrations (referee#2's concerns) may be easily ignored possibly due to unnoticeable locations. For example, it is noted that simulation results shown herein are concluded based on eighteen cases covering severe regional drought processes over all the climatic regions over mainland China during 1981-2016, most of which are displayed in the supplement files. Anyhow, we will try to illustrate them and make it as clear as possible.

Last but important, results in the paper generally consists of three parts: 1) relevant physical backgrounds in the form of routine climate diagnose on evolutions of vertical motion/horizontal divergence; 2) a newly proposed SVVAI and its application in simulating drought processes; 3) new attempt of drought prediction using the forecasted SVVAI. The first two parts are our major goals, and the third one is preliminary and displays the potential of being applied to drought prediction. It is a little difficult to solve all concerns in one single paper.

Major comments

1.     The English is not of the required standard. It is frequently a significant challenge to interpret the text, requiring very careful reading. This makes it very difficult to focus on the intended presentation and interpretation of results. I suggest the authors improve the grammar before resubmission.

Response:

As commonly pointed out by Referee#1 and #3, English expression herein needs to be improved before resubmission. We will surely self-check, ask help from native English speakers to improve it.

2.     The conclusions for both drought simulation and prediction are hampered by sample size. By selecting only a few case studies, it is difficult to draw overall conclusions about the relative benefits of SPI3 versus SVVAI. For example, how do the two indices compare during (a) other droughts, and (b) on all days (not just droughts)?

Furthermore, the authors' conclusions about the SVVAI being equal or superior to the SPI3 in predictive performance (L316) to be overstated. With only a small sample of events and no use of forecast verification metrics, it is impossible to draw this kind of conclusion.

Response:

As climate extreme events, severe drought processes rarely happen, leading to small sample sizes in nature. Even so, the severe drought events without seasonality consideration during 1981-2016 over all the climatic regions in China are research targets herein, which tend to result in relative general conclusions. In this situation, sample sizes herein are not so small.

However, we agree on the comment of making comparisons on all days (not just droughts). It is an important test whether the simulated SVVAI systematically corresponds to SPI, especially two tails (severe droughts and pluvial). We will add relevant illustration in the discussion part and try to implement it in our further researches.

We disagreed with the comment concerning the conclusions near L316. Preliminary comparisons of drought prediction between the SVVAI and SPI were conducted based on RAW forecast products indeed. We understand post-process procedures of hydrometeorological variables are important for operational forecasting. However, comparisons made based on RAW products are also necessary especially for a preliminary exploration of drought prediction using SVVAI. Actually, that further and specified post-process procedures are necessary were mentioned in LINEs 423-426 (in the discussion part).

Another concern from Referee#2 seems to be no use of forecast verification. Quantitative assessments (Fig.8 and Fig. S10) based on the mean and standard deviation of sample bias (see LINE 316-334) is one form of forecast verification metrics. This is a little different from routine forecast validations because process-based forecasting is our concern.

3. I do not think the authors adequately describe the reasoning behind using the hor-izontal divergence and vertical velocity as diagnostics of drought. Dynamically, how do we expect these variables to change during drought? Is that seen in the results presented here (e.g. Fig 2). The paper could do more to link the results to dynamical processes. For me this is a "major comment" because it underpins the whole point of the paper.

Response:

Dynamical subsidence is one important drought-inducing factor based on general knowledge of atmospheric science. To explain it as clearly as possible, we have a detailed introduction in LINEs 66-85, a very specified analysis of case studies in section 3. Dynamical process diagnoses during the drought (Figure 2 and Figure S2) and associated vertical profiles as a function of different periods (Figure 3 and Figure s3) are important backgrounds of the newly proposed SVVAI indeed.

Referee#2's comments say the paper could do more to link results to dynamical processes. What we illustrated above is what we can do. To tell the truth, we have no better ideas regarding this comment, but we will think about it deeply.

4. There should be greater discussion of the relative benefits of SVVAI and SPI3. Why would you use SVVAI when you could just use SPI? In what circumstances is it preferable to use SVVAI? Can you draw those conclusions from the results presented here?

For example, Fig 6 implies that SVVAI is only comparable to SPI3 for eastern China, but there is no discussion of this. Does this mean SVVAI is not a useful indicator of meteorological drought for the rest of China?

Response:

Discussion about relative benefits and preferable application of SVVAI can be found in LINEs 396—401 (the third issue in the discussion part). We treat SVVAI as a complementary index compared to SPI, and we agree to confirm circumstances in which SVVAI is preferable.

Regarding the discussion about why SVVAI is comparable to SPI3 for eastern China shown in Fig.6, detailed climate-related illustration can be found in LINEs 292—299.
Performance of SVVAI in Figure 6 help understand why drought simulation in eastern China perform better (section 4.1 and 4.2). However, it did not mean it is not useful for other droughts for the rest of China, please see performance regarding droughts in regions except eastern China in section 2 and 3.

5. There is too much information presented across all the figures. I think some of the information should be synthesised. For example, is it really necessary to have 3 similar figures, each with 11 panels (Figs 9, 10, 11)? Surely there are ways of presenting this information more concisely. I also do not advocate simply moving some figures to the supporting information, as that already contains many results.
Response:
Figs 9-11 correspond to the 2011 mega-drought with a spatially huge coverage (illustrated in LINEs 340—343), which is a good case to verify the application of the SVVAI at the grid-scale nationwide. However, we think this comment is also reasonable, and we will further move one or two figures into the supplement files for simplification.

6. I find the conclusions of the forecasting section to not match the results. As mentioned in point 2, the statements about predictive performance are severely hampered by the sample size of events and lack of forecast verification metrics.
Response:
The corresponding response can be found in point 2.

Greater synthesis of the forecast results is needed. For example, for Figs 9, 10 and 11, the authors draw conclusions about model performance by visual inspection of spatial fields. While it is useful to visualise it this way, I do not think it needs to be done 3 times over (this is related to point 5). In fact, the authors already use a measure of pattern correlation (PCC). Could PCC not be used to quantify the similarity of forecast fields to the observation, or between the two models' forecast fields, as a way of presenting results more generally?
Response:
Pattern correlation coefficients (PCC), which is so-called Anomaly Correlations (AC), can surely be used to quantify the similarity of forecast fields to observation. Details about PCC (i.e., Anomaly Correlation) in section 8.6.4 of the book (Wilks, 2011). Anyway, we will move the key illustration and references from supplement files to section 2.6 to make it clear.
References:
Wilks, D. S. (2011). Statistical Methods in the Atmospheric Sciences (3 ed. Vol. 100): Academic Press.

Furthermore, I am concerned by the lack of forecast calibration. To be fair, the authors note that the CFSv2 forecasts used in SVVAI prediction are raw (L423-426), and could benefit from calibrating to the ERA-Interim vertical velocity and horizontal divergence data. However, it is not mentioned that the precipitation forecasts are also raw, and hence the SPI3 also require post-processing to indicate operational forecast skill. It is entirely possible (probable, even?) that the biases in the precipitation forecasts are larger than the biases in the atmospheric variables (in fact I think this is what the authors are saying in L83-85). Therefore, the forecast results are presented with the models on an unequal footing, making comparisons very difficult.
Response:

To tell the truth, this comment might be reasonable but a little bit confusing. We derived precipitation and vertical motions featuring omega from raw CFSv2 forecast products and treat them on an equal footing. Since forecast precipitation might have experienced post-processing procedures before release (means that precipitation products have better qualities than omega products), the promising performance of SVVAI in drought prediction may be more convictive.

Minor comments

1. Figure captions are not detailed enough. For example, what is Figure 7 showing?Why the many different orange and black lines? I assume they are different initialisation dates but please describe what's shown in the figures.

Response:

Thank you for the comment, as also pointed out by Referee#1. We will surely check figure captions and make them detailed and clear.

2.    L153: What does the "critical proportion" mean? Please explain. Is this where you define "extreme" and "severe" drought processes?

Response:

Yes, it is related to the way we define grades of drought processes. We will make the description in section 2.3 clear. Thank you for the comment.

3.    Figure 2 caption: "shadow" should be "shading".

Response:

We will modify it.

4.    Figure 4, 7: The y-axis label should surely be something like "Standardized anomaly" rather than SPI3?

Response:

Some of them will be revised as SVVAI.

5.    Section 6: Remind readers which forecast model is used (CFSv2) at the beginning of the section.

Response:

We described it in section 2.6. Anyhow, we will surely mention it again at the beginning of section 6.

6.    L316: How do the authors come to the conclusion that SVVAI_ave is equal or superior to SPI3? Please walk the reader through the figure so that they can draw the same conclusions.

Response:

We agree with the comment and will revise it. Originally conclusions that SVVAI_ave is equal or even superior to SPI were drawn based on Figure 8 and Figure S10. We think it reasonable but will weaken/limit the conclusion because raw products are employed without hydrometeorological post-processing procedures indeed.

---

## Author Comment (AC3) · 20 Jul 2020

**Response to comments from Referee#3:**

This manuscript discusses the dynamics of droughts over different climate regimes in China. The authors depart from the many typical drought studies that are based on sur-face indices, offering a new and refreshing approach based on atmospheric dynamical principles. They propose to use vertical velocity and convergence/divergence patterns to define two new drought indices. I find the concept exciting and promising.

The study uses global reanalysis data to assess the general behavior of the proposed index against the traditional approach of using SPI. Then, they employ long-term operational forecasts to evaluate whether the new indices contain a predictive signal, in which case they could be used as a drought forecast tool.

The analysis is carried over several case studies and thus cannot offer a statistical significance. Some regions of China towards the east seem to respond consistently to the dynamical hypothesis, but this is not the case of droughts in other areas towards the west. The authors are honest and cautious about the possible limitations of the approach. In this reviewer's view, the study is a promising approach that has the potential to complement traditional drought studies. In that sense, I would not take the results as a definitive answer but as the starting point for other studies in this matter.

I would argue that the manuscript meets most of the review criteria for this journal, as defined in https://www.earth-system-dynamics.net/peer_review/review_criteria.html, with one exception. The manuscript is, in the most part, understandable despite limitations with the language and grammar. My suggestion to the authors is to get help from an editorial office or native English speaker that can review and help correct the grammar. They could also use software like Grammarly that helps detect and offer suggestions to many of the weakly formed sentences. Grammarly, and likely other equivalent software, offer subscribers additional support from an expert team at a fee.

My recommendation is that the manuscript should be published after those corrections. It would be a loss if poor grammar were used as the main factor to prevent publishing.

Response:

Thank you for your understanding and the comments highlighting the originality of this paper. It is a new attempt but not perfect. Some further studies have been conducted. This paper tends to be footstones for other new attempts in drought communities because it indicates that the evolution of dynamical subsidence coincides with drought development and can be generalized as feasible drought indicators.

As commented by Referee#1 and #2, there are some shortcomings indeed. However, it is hard for us to solve all the concerns in one single paper. We will improve it in our further study.

We will surely improve relatively poor languages and grammars using software called Grammarly and then seek help from native English speakers.